# Cognitive fatigability and neuronal correlates in chronic pain – A cross-sectional fMRI study

Anna Holmqvist[1,2]*, Love Engström Nordin[3,4], Nils Berginström[5,6], Monika Löfgren[1,2], Lars Nyberg[7,8], Britt-Marie Stålnacke[5], Marika C. Möller[1,2]

1 Department of Clinical Sciences, Karolinska Institutet, Stockholm, Sweden, 2 Department of Rehabilitation Medicine, Danderyd University Hospital, Stockholm, Sweden, 3 Department of Neurobiology, Care Sciences and Society (NVS), Karolinska Institutet, Stockholm, Sweden, 4 Department of Diagnostic Medical Physics, Karolinska University Hospital, Solna, Stockholm, Sweden, 5 Department of Community Medicine and Rehabilitation, Rehabilitation Medicine, Umeå University, Umeå, Sweden, 6 Department of Psychology, Umeå University, Umeå, Sweden, 7 Department of Diagnostics and Intervention, University Hospital of Umeå, Sweden, 8 Department of Medical and Translational Biology, Umeå University, Umeå, Sweden

* anna.holmqvist@ki.se

## Abstract

### Objectives

Fatigue is common in patients with chronic pain. However, there is a knowledge gap concerning performance fatigue, cognitive fatigability, and its neural correlates in this patient group. In this study, we therefore aimed to investigate the presence of cognitive fatigability and its neural correlates in patients with chronic pain using functional magnetic resonance imaging (fMRI).

### Methods

In this study 24 women with chronic pain and 22 healthy controls, aged 18–45 years, underwent a 20-minute psychomotor vigilance task (PVT) assessing reaction time during blood-oxygen-level dependent (BOLD) fMRI. Reaction time and BOLD signal changes were analyzed using dual regression.

### Results

The patients showed significant cognitive fatigability, i.e., prolonged reaction time, during the PVT, while the controls maintained a stable performance ($p = .018$). No significant neural time on task effect was found on BOLD fMRI. There were however differences in the brain areas activated between the groups throughout task performance. Patients with chronic pain showed stronger activation primarily in prefrontal cortex, including motor areas, while healthy controls demonstrated stronger activation particularly in the left middle orbital gyrus and right insula.

**Data availability statement:** The data studied in this paper is available from https://doi.org/10.5281/zenodo.17341991, and can also be found in the link in our Supporting Information file.

**Funding:** Stiftelsen Promobilia (no. A22056). The Department of Clinical Sciences, Karolinska Institutet. Regional agreement between Umeå University and Västerbotten County Council on cooperation in the field of Medicine, Odontology, and Health. The funders had no role in the study design, data collection and analysis, decision to publish, or preparation of the manuscript.

**Competing interests:** The authors have declared that no competing interests exist.

## Conclusion

The study indicates that the phenomenon of cognitive fatigability is present in patients with chronic pain. Also, the presence of chronic pain was associated with increased activation in brain regions related to motor and cognitive control, possibly reflecting compensatory mechanisms. Conversely, healthy controls showed higher activity in regions active in motivation, reward, and decision-making, suggesting more motivation-driven and efficient processing.

Further studies are needed to validate the results.

## Introduction

Fatigue is a common symptom in patients with chronic pain, and has traditionally been measured as a subjective experience, using self-report scales [1]. Experience of fatigue can be categorized into different subtypes, such as physical or mental, but also cognitive fatigue, the latter being conceived as the experience of fatigue resulting from cognitive effort [2]. Cognitive fatigue has, however, not been well studied in patients with chronic pain [3]. An aspect of cognitive fatigue is cognitive fatigability (CF), which means that cognitive performance deteriorates over time. Thus, CF can be assessed objectively as performance decrement on tasks requiring sustained mental effort [4,5]. Studies on patients with neurological conditions have demonstrated that CF can be induced during a short period (a few minutes) of effortful performance, provided that the task is sufficiently cognitively demanding [6,7].

An alternative way to an objective assessment of cognitive fatigue and CF is to use functional magnetic resonance imaging (fMRI) methods to capture alterations in brain connectivity during task performance. It has been suggested that fMRI might be a more appropriate approach to objectively mirror subjective fatigue as compared to performance-based measurements [8,9] where research historically has found sparse correlation [7].

Fatigue in acquired brain injury has been explained by increased cerebral activity in the injured brain, serving as a compensatory mechanism for slowed processing speed and attention deficits [10]. Also, previous fMRI studies on fatigue have shown an association between self-reported fatigue and patterns of neural activity during the performance of cognitively demanding tasks, partly differing between patients with acquired brain injury and neurological conditions and healthy subjects. The impacted regions are related to attention and executive control, including the basal ganglia, prefrontal cortex (PFC), thalamus, anterior cingulate cortex (ACC) and superior parietal areas [5,9,11–13]. Chaudhuri and Behan, in their widely cited model, also put emphasis on deficits in motivation, due to dysfunction within the striatal-thalamic-frontal cortical network, as crucial to centrally mediated fatigue [14,15]. In line with this, it has been suggested that subjective fatigue might result from an imbalance between effort and reward [16].

Cognitive disturbances, including reduced processing speed and attention deficits, are common in chronic pain patients [17,18], presumably due to signals of pain

competing for a share of limited attentional capacity [19,20]. Thus, it is plausible that similar mechanisms may underlie cognitive fatigue and possibly CF in chronic pain as in brain injury and neurological conditions [21]. There is to date a paucity of studies targeting objective aspects of cognitive fatigue in patients with chronic pain, though one study has shown an association between CF and processing speed in the patient group [3].

To shed further light on this subject, in this study we aimed to investigate the presence of CF during a 20-minute vigilance task in patients with chronic pain, and to explore whether there is a difference in blood oxygen level dependent (BOLD) signal during the performance of the task between patients with chronic pain and healthy subjects.

## Materials and methods

The present study, which has across-sectional design, is a sub-study of a larger clinical trial (ClinicalTrials.gov NCT05452915). Detailed information on the measures can be obtained from the study protocol [22]. Power calculation was based on an earlier study on mild traumatic brain injury [13]. The study is presented according to the STROBE guidelines.

### Participants

Twenty-four right-handed female patients, aged 18–45 years with chronic pain, were compared to a convenience sample of 22 healthy controls.

### Inclusion criteria

Chronic pain, duration > 3 months, according to the International Association for the Study of Pain definition (IASP) [23]; age 18–45 years; referral for team assessment due to chronic pain. Since the vast majority of the patients in the main study were female [3], solely women were included in this sub-study, to avoid confounding due to gender differences.

### Exclusion criteria

Acquired brain injury (including concussion); severe psychiatric disorder; intellectual disability; medication potentially affecting cognitive functions (e.g., sedatives, opioids); pregnancy; non-fluency in the Swedish language; implants constituting a contraindication for MRI; fear of confined spaces; left-handedness. As to the healthy controls, in addition, chronic or present pain were exclusion criteria.

### Recruitment

All participants underwent team assessment at the Pain Center at Umeå University Hospital from September 1, 2020 to September 30, 2021. Those willing to participate in the fMRI study and who fulfilled the criteria for participation were consecutively included. The control group was a convenience sample, recruited through billposting at the University Hospital of Umeå. Out of 39 healthy persons reporting interest to participate in the study, 22 were included to closely match the patient sample in age and educational background at the group level. The controls were included in parallel with the patients, and care was taken to ensure that gender, age and length of education matched the patient group. However, it was difficult to recruit controls with a low level of education, resulting in an incomplete match in educational background.

### Instruments and procedure

**Measurements.** Background information (Table 1) was obtained from the Swedish Quality Register for Pain Rehabilitation [24]. Healthy controls provided this information in adjacent to the neuropsychological examination.

**Table 1. Clinical and demographic data for patients and controls.**

| | Patients n=24 | Control group n=22 | p-value |
|---|---|---|---|
| Age years | 33.4 (7.9) | 29.8 (6.4) | 0.100 |
| Female, n (%) | 24 (100%) | 22 (100%) | N/A |
| Education years | 13.5 (2.0) | 15.2 (1.6) | 0.003 |
| Matrix reasoning | 18.5 (4.3) | 19.4 (3.6) | 0.448 |
| Pain duration years | 8.3 (6.0) | N/A | |
| Spreading of pain 0–36 | 18.3 (8.6) | N/A | |
| VAS pain before MRI session 0–100 | 42.9 (17.7) | 2.6 (6.5) | <0.001 |
| VAS pain after MRI session 0–100 | 56.5 (22.7) | 3.6 (7.1) | <0.001 |

Note: No missing data. Data are means (standard deviations), counts (percentages).

Abbreviations: MRI – Magnetic Resonance Imaging; N/A – non applicable; VAS – Visual Analog Scale

fMRI task: Behavioral data.

## Outcome measure Fatigability

**Psychomotor Vigilance Task (PVT).** The PVT [25] was designed using e-prime software (Psychology Software Tools Inc., Pittsburg, PA) as a sustained attention task. The task has been shown to be reliable and sensitive for assessing fatigue-dependent attentional and performance deficits [25].The PVT session within the MRI scanner lasted for 20 minutes. The test setup comprised a projector-based visual system, equipped with feed-back mechanisms to measure the participants´ reaction times (RT). All MRI images were examined by a radiologist.

The participants were instructed to press a button as quickly as possible upon seeing a set of four zeroes within in a red rectangle, and to do nothing if other numbers appeared. Following each response, visual feedback of the reaction time was presented for 1 second. If the participant pressed the button at a false stimulus or if the response time exceeded 1 second, the feedback displayed "false answer" or "no answer" respectively. The intervals between stimuli varied from 2 to 10 seconds in a pseudo-random manner. Although the number of true positive responses varied slightly among individuals (ranging from 170–210 responses) the entire task always lasted 20 minutes.

The number of RT responses from the PVT was categorized into quartiles, with each quartile containing an equal number of responses for each individual. The mean response time was given for each quartile. However, as the number of responses within quartiles could vary among individuals due to the random distribution of stimuli, the quartile division is less precise in time compared to the BOLD fMRI data. CF was defined as a negative value when the mean reaction time of the last quartile of the PVT was subtracted from the mean reaction time of the first quartile.

## Demographic variables

**Visual analog scale of pain (VAS pain).** Present pain was assessed using a 100 mm VAS ranging from 0 (no pain) to 100 (worst pain imaginable).

**Spreading of pain.** The spreading of pain was assessed by counting the number of painful sites on the body. The assessment was based on 36 predefined anatomical areas covering the four quadrants of the body, 18 on the front and 18 on the back [24].

**Matrix reasoning.** Matrix reasoning [26] assesses non-verbal logical reasoning and is considered robust against cognitive decline. Higher scores indicate better performance. This test was used to compare potential differences in premorbid level between patients and the control group.

**Procedure.** Prior to examination the participants were provided with written and verbal information about the study and subsequently signed informed consent. Following blood sample collection (not reported in this study), they were offered

breakfast, including tea or coffee, before undergoing MRI examinations. Before and after the MRI session the participants were asked to fill in a form, including a rating of current pain. The MRI sessions started at 9:00 and lasted approximately one hour. Before the MRI scans, there was a 2-minute training session of the PVT.

All subjects underwent neuropsychological assessment, either before or after the MRI scanning session, though not on the same day as the scanning. In this study only the results from the Matrix reasoning test have been analyzed.

**MRI procedure.** The MRI examination included an initial resting state session, task fMRI, a second resting state session and structural image acquisition. The image acquisition was performed using a 3T GE Discovery MR 750 (General Electric, Milwaukee, Wisconsin) equipped with a 32 channel phased array head coil. During the PVT a T2-weighted single shot echo planar imaging sequence was acquired for BOLD fMRI contrast. A repetition time of 2000 ms and echo time of 30 ms was used. Slice thickness was 3.4 mm with a slice gap of 0.5 mm and interleaved slice acquisition order. The flip angle was 80°, field of view 250x250 mm and matrix 128x128. The sequence lasted for 20 minutes with 600 volumes. The sequence protocol also included resting state fMRI before- and after the PVT using the same sequence but with 170 volumes per measurement. The resting state fMRI results will be presented in a separate paper. Except for the BOLD fMRI sequences, a T1-weighted structural image and a T2 weighted FLAIR sequence was acquired for clinical evaluation of the study participants.

**Image post-processing.** The BOLD fMRI data was preprocessed using FSL (http://www.fmrib.ox.ac.uk/fsl). Intra-modal motion correction was performed using McFLIRT, by applying rigid body transformations to correct fMRI data for motion [27]. The time curves for translational and rotational motion were assessed for all participants to make sure that head motion did not exceed 1 voxel size (2 mm). Data with motion exceeding 2–3 mm cannot be reliably corrected with standard motion correction algorithms and can therefore cause misregistration. Brain extraction tool, BET, was used to remove non-brain tissue [28]. Spatial smoothing was used with a 5 mm full width at half maximum (FWHM) to minimize noise. Spatial normalization to the standard Montreal Neurological Institute brain template was performed using a 12-parameter affine transformation and mutual-information cost function. During the affine transformation the imaging data was re-sampled to isotropic resolution of 4 mm.

Since the PVT used a random design, it is not trivial to test the time course of each voxel against a hypothesized wave-form. Instead, independent component analysis, ICA, was used. ICA is an exploratory technique which is effective in denoising random noise and confounding signals from physiological motion, and it allows for detection of random responses [29]. The ICA analysis in the present study was performed using the FSL tool MELODIC version 3.15 and the multisession temporal concatenation approach [30]. ICA was performed on all participants to obtain good statistics for the independent components representing all participants in the study and thereby avoiding any group bias. A dual regression analysis was used to map the group level components back to individual subjects data to analyze time effects and within-group differences. To do this, the spatial maps from the ICA were used to generate subject-specific versions of the spatial maps, and associated time series, using dual regression [31]. First, for each subject, the group-average set of spatial maps is regressed into the subject's 4D space–time dataset. This results in a set of subject-specific time series, one per group-level spatial map. Next, those time series were regressed into the same 4D dataset, resulting in a set of subject-specific spatial maps, one per group-level spatial map. We tested for group differences using FSL's randomize permutation-testing tool [32]. To correct for multiple comparisons in the group analysis, 10000 permutations were performed using randomize resulting in each single voxel being significant at a p-value of.05. Contrast matrices were created in FSL's general linear model tool, GLM, to perform two-way analysis of group effects. With the results we could assess group differences in the functional connectivity of brain networks. To investigate time effects, using dual regression, each data set was divided into quartiles including 150 volumes per time point. Separate contrast matrixes were constructed using GLM to search for within group time effects by comparing each quartile to all other quartiles. This analysis was performed in both directions for patients and controls respectively. Also, the time effect analyses were corrected for multiple comparisons using 10000 permutations with randomize and a Bonferroni corrected p-value. No data sets were excluded because of motion.

### Statistics behavioral data

Parametric methods were applied for the vigilance test, as data showed to be normally distributed. Independent Student's *t*-test and Repeated measures ANOVA were used for comparison between patients and controls. As to the latter, Mauchly's test for sphericity being significant, the Greenhouse-Geisser correction was performed. Two-tailed *p*-values were used with a critical significance level of $p < .05$. Data were analyzed in IBM SPSS, version 22.

### Ethical approval

The study was approved by the Swedish Ethical Review Authority (2018/424–31; 2018/1235-32; 2018/2395-32; 2019−06148; 2022-02838-02) and performed according to the declaration of Helsinki. Informed consent was obtained from all paricipants in the study.

## Results

### Demographics

There were no differences in age or estimated premorbid intellectual level (Matrix reasoning) between the chronic pain group and the control group. However, the control group had a significantly higher educational level. The chronic pain group had a mean duration of pain of 8.3 years (Table 1). Most of the patients suffered from musculoskeletal pain conditions.

### PVT performance

There was no mean difference in RT on the PVT between the groups (Table 2) and no significant correlation between total RT and self-rated pain, nor between CF (difference between first and forth quartile) and self-rated pain in either group. Nor did level of education correlate significantly with total RT or CF.

One way ANOVA for repeated measures was conducted to investigate if time on task affected changes in RT during the PVT within subjects (time effect) and between groups (time and group effect. The result showed a time on task effect of the RT $F_{(2.05, 90.02)} = 7.107$ ($p < .001$), but also a significant time vs group interaction effect $F_{(2.05, 90.02)} = 4.165$ ($p = .018$), as the patients increased their RT during the task while the controls had a stable reaction time (Fig 1).

When searching for changes in BOLD fMRI signal between different quartiles, we used corrected p-value threshold of 0.025. We also applied a p-value threshold of 0.05 to see if there were any weak effects or trends in the data. We were

**Table 2. Values (milli seconds) of the Psychomotor Vigilance Task (PVT).**

|  | Patients, n = 24 | Controls, n = 22 | p-value |
|---|---|---|---|
| Total Mean RT PVT, m (sd) | 395. 68 (80.89) | 360.33 (42.35) | .069 |
| Mean RT PVT Q1, m (sd) | 381.49 (58.92) | 363.52 (34.88) | .212 |
| Mean RT PVT Q2, m (sd) | 387.89 (83.24) | 354.60 (42.21) | .092 |
| Mean RT PVT Q3, m (sd) | 400.21 (90.64) | 358.77 (48.99) | .059 |
| Mean RT PVT Q4, m (sd) | 415.28 (96.88) | 365.68 (54.03) | .037 |
| PVT fatigability, m (sd) | 33.79 (48.12) | 2.16 (35.18) | .015 |

Note: No missing data. Data are mean and standard deviations.

Abbreviations: m – mean; PVT – Psychomotor Vigilance Task; Q – Quartile; RT – Reaction Time; sd – standard deviation

There were no significant correlations between current pain measurements (VAS pain) and the results of the PVT.

fMRI data.

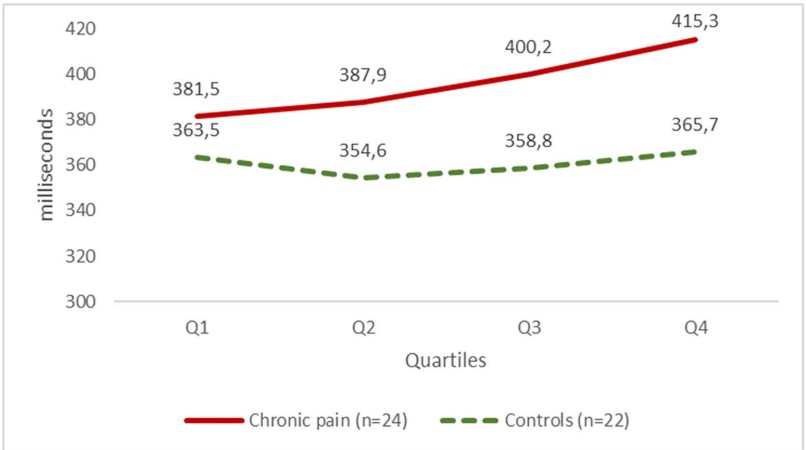

**Fig 1. Fatigability in patients with chronic pain and controls as measured by the PVT.** Fatigability values equal the mean scores of the reaction time in each quartile. Abbreviations: CP – chronic pain; CG – control group; PVT – Psychomotor Vigilance Task; Q – Quartile.

not able to find any significant time effects or trends. However, we found differences in activated brain areas between the groups throughout task performance. The patients showed greater connectivity foremost in left superior frontal gyrus and right precentral gyrus compared to the controls and lower activity than the controls in several areas, most notably in the left middle orbital gyrus. For details, see Table 3 and Fig 2.

## Discussion

In this study, we aimed at investigating the presence of CF during a 20-minute vigilance task in patients with chronic pain, and to explore differences in BOLD signal between patients and healthy controls during task performance.

The result showed significant CF in patients with chronic pain while the controls showed a stable performance. We also found differences in the BOLD fMRI signal between the groups, however we found no effect of time nor an interaction effect of group vs time in the BOLD fMRI signal when comparing regional blood flow during the different quartiles of the vigilance task. While we observed no significant time and group dependent effects, there were group differences in the patterns of brain activation throughout task performance.

The CF seen in the patient group goes in line with what has previously been showed in studies with patients suffering from neurological conditions and brain injury [6,7] and gives support for the notion that patients with chronic pain, alike, seem to be more prone to CF than healthy subjects. Interestingly, the present results contrasts to that seen in a study newly performed by our group [3]. In the previous study we did not find more CF in patients with chronic pain as compared to healthy controls during the performance of the cognitively demanding Coding task, which has been shown to elicit CF in patients with acquired brain injury [6]. As the Coding task is of shorter duration (2 minutes) than the PVT, it is possible that task length might be a crucial/differentiating factor. Wang et al. and Holtzer et al. [33,34] have shown that extended periods of cognitive work, not less than 30 minutes, is warranted to induce performance decrements in healthy subjects, and it is possible that this, to some extent, also holds for patients with chronic pain. Furthermore, these results indicate that task length might outweigh task complexity in eliciting CF in patients with chronic pain, the vigilance task being a relatively simple reaction time test, though demanding constant alertness [25], as compared to the executively demanding Coding task [35].

Additionally, a differentiating factor between Coding and the PVT concerns task constraints. While the Coding task permits freedom to regulate performance speed, the vigilance task imposes a fixed pace, which logically would increase the

**Table 3. Results from group analysis with dual regression based on the MNI brain template. A *p*-value threshold of 5% has been used meaning that all clusters in the table are significant at p 0.05. The MNI coordinates represents the centre point of each cluster.**

| Brain region | IC | MNI coordinate(x, y, z) | Cluster volume (cm³)/ # voxels |
|---|---|---|---|
| Regions with significantly greater connectivity to group level components for patients compared to healthy controls | | | |
| Left Inferior Frontal Gyrus | 2 | −46, 42, −8 | 0.128/ 10 |
| Right Precentral Gyrus | 14 | 26, −14, 56 | 0.448/ 35 |
| Left Paracentral Lobule | 14 | −6, −22, 64 | 0.320/ 25 |
| Left Precentral Gyrus | 14 | −14, −18, 68 | 0.128/ 10 |
| Left Superior Frontal Gyrus | 15 | −26, −10, 60 | 0.960/ 74 |
| Regions with significantly greater connectivity to group level components for healthy controls compared to patients | | | |
| Left Posterior cingulate cortex | 7 | −10, −46, 28 | 0.320/ 25 |
| Within 1 mm Left Precuneus | 7 | −14, −50, 16 | 0.128/ 10 |
| Right Insula Lobe | 19 | 34, 22, 12 | 2.240/ 173 |
| Left Supplementary Motor Area | 19 | −2, 14, 60 | 1.280/ 99 |
| Left Cuneus, within 1 mm Left Precuneus* | 19 | −10, −62, 28 | 1.024/ 79 |
| Left Inferior Temporal Gyrus | 19 | −50, −22, −20 | 0.896/ 69 |
| Right Supplementary Motor Area | 19 | 10, −18, 72 | 0.832/ 64 |
| Right Middle Frontal Gyrus | 19 | 34, 38, 24 | 0.768/ 59 |
| Within 4 mm Right Insula Lobe | 19 | 30, 34, 8 | 0.640/ 49 |
| Within 1 mm Left Hippocampus* | 19 | −18, −34, 8 | 0.448/ 35 |
| Right Thalamus* | 19 | 10, −18, 0 | 0.128/ 10 |
| Left Middle Orbital Gyrus* | 20 | −6, 54, −12 | 8.768/ 676 |

*Overlap between significant cluster and IC.

Abbreviations: IC – Independent Component; MNI – Montreal Neurological Institute.

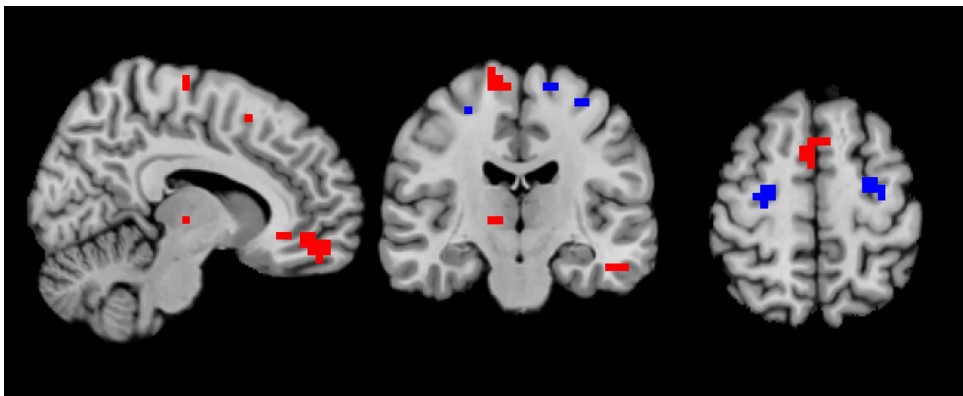

**Fig 2. A depiction of areas significantly more activated in patients (blue) and controls (red) respectively during the psychomotor vigilance task performance.**

likelihood of CF. In support of this theory, the patients were significantly slower than the controls on the Coding task in our previous study [3], which might indicate that they compensated their CF with a generally slower performance.

Another objective of the study was to investigate the neuronal correlates of CF in patients with chronic pain.

The lack of a time-dependent effect in BOLD fMRI signal in this study contrasts with previous findings in neurological conditions, where time-on-task has been shown to alter activation patterns within the striatal-thalamic-frontal cortical network, differing between patients and healthy subjects [5,11]. A reason for this might be the strict significance thresholds applied in this study to avoid false positive findings. A larger sample might have been more sensitive to detect differences in perfusion over time between the groups.

While we observed no significant time and group dependent effects, there were group differences in the patterns of brain activation throughout performance on the PVT task. A brain network primarily comprising somatosensory cortical areas (S1 and S2), insular cortex, thalamus, ACC and PFC has been identified to be active in the processing of noxious stimulation [36], and associated with perceived pain intensity [37]. In chronic pain conditions, though, somatosensory cortex and limbic structures (insular cortex and ACC) seem to decrease in activation, along with an increasing activity in PFC [36]. This picture goes in line with the patient group showing increased activation in left superior frontal gyrus, which is part of the dorsolateral PFC, considered crucial for higher-order cognitive functions, e.g., planning, cognitive flexibility and working memory [38]. Also, left superior frontal gyrus is located in the pre supplementary motor area (pre SMA), a region shown to be active in modulation of effort [33], response selection and inhibition [39]. In the context of the present study, the finding might reflect the mere processing of chronic pain in the patient group, but also a need to recruit greater PFC resources to sustain performance despite intrusive pain signals.

Furthermore, the patient group showed increased activity in frontal areas primarily related to motor function and motor control, not least in right precentral gyrus. An explanation for this finding might be that the vigilance task includes a fine motoric element, i.e., a requirement of intermittently pressing a button, which might have been harder to execute for the patients than for controls. Patients with fibromyalgia, a condition characterized of widespread chronic pain, have been shown to have deficits in motor control and dexterity [40]. It is possible that the patient group, whereof the majority showed pain in multiple pain sites, needed to allocate more resources on the motoric part of the task than the healthy controls to deal with pain-induced motor disturbances.

However, as these regions are included in IC 14, which has a strong white matter involvement, suggesting it is not a purely functional network, significant group differences need to be interpreted with caution. There was, though, no difference in white matter integrity between the patients and controls. The patients showed less activity than healthy controls in several brain areas, predominantly in the left middle orbital gyrus. Left middle orbital gyrus is part of the orbitofrontal cortex which is considered to be involved in decision-making and expected reward-value [41]. Cognitive fatigue in patients with neurological disorders has been attributed to decline in motivation due to imbalanced effort-reward calculation, caused by malfunctions in the fronto-striatal network [2,16,42]. Our findings suggest that this may also apply to patients with chronic pain, i.e., lower activity in the orbital gyrus meaning reduced sensitivity to rewards. This decreased sensitivity might, consequently, lead to hampered motivation, ultimately resulting in cognitive fatigue, and according to our findings, possibly also CF. This reasoning might as well tentatively apply to the reduced activation observed in the insula within the patient group, consistent with Apkarian´s et al. [36] findings of decreased insular activation as pain becomes chronic. This since the insula is a brain region crucial for interoceptive awareness and emotional processing [43], and furthermore has been suggested to be included in a motivational fatigue network [2].

A limitation of the study was the inclusion of women only, and in a restricted age span, with the upper limit set at 45 years to avoid effects of age-related neurocognitive decline. This prevents generalization to the male population and to other age-groups. However, the examined groups were homogenous, and confounding factors such as history of brain injury, substance abuse and psychiatric illness were controlled for by use of strict exclusion criteria.

Also, the control group had significantly higher levels of education, which may have influenced both cognitive performance and patterns of brain activation.

Another constraint concerns the study design, which strictly does not permit conclusions to be drawn on whether the observed differences in brain activation between the groups is an effect of variations in the processing of effortful cognitive

work between subjects with chronic pain and healthy controls, or an effect of the mere presence of chronic pain. However, we found no correlation between current pain intensity and PVT- performance.

To be noted, the role of depression was not accounted for in this study; though previous studies have found no correlation between depression and CF, as opposed to CF and subjective fatigue [6,44].

In conclusion, this study is to our knowledge the first to demonstrate the phenomenon of CF not only in neurological conditions and brain injury, but also in patients with chronic pain.

Chronic pain was associated with increased activation primarily in frontal brain areas related to motor control and cognitive effort, reflecting potential compensatory mechanisms. Lower activity in the patient group was seen in areas associated to expected reward-value, indicating impaired reward-processing in patients with chronic pain. These findings align with existing literature [21,45,46], suggesting that chronic pain leads to widespread brain function changes, affecting cognitive, emotional, and motor functions.

The results might be of clinical relevance. The indication of impaired reward-processing in patients with chronic pain opens for the opportunity to study interventions targeting motivation, including neurofeedback, cognitive training, or pharmacological interventions, to reduce cognitive fatigue and CF associated with chronic pain. However, the lack of time-dependent neural effects despite behavioral fatigability weakens the link between brain activity and CF. It should be stated that this is a small explorative study, and that repeated well-designed studies that could confirm our preliminary findings are required.

## Supporting information

**S1 File. Link for access fMRI data.**
(DOCX)

## Acknowledgments

The authors would like to thank the clinical staff for assisting in the recruitment of patients.

## Author contributions

**Conceptualization:** Anna Holmqvist, Love Engström Nordin, Nils Berginström, Monika Löfgren, Lars Nyberg, Britt-Marie Stålnacke, Marika C Möller.

**Data curation:** Anna Holmqvist, Love Engström Nordin, Nils Berginström, Lars Nyberg, Marika C Möller.

**Formal analysis:** Anna Holmqvist, Love Engström Nordin, Lars Nyberg, Marika C Möller.

**Funding acquisition:** Anna Holmqvist, Nils Berginström, Monika Löfgren, Britt-Marie Stålnacke, Marika C Möller.

**Investigation:** Nils Berginström.

**Methodology:** Anna Holmqvist, Love Engström Nordin, Nils Berginström, Lars Nyberg, Marika C Möller.

**Project administration:** Anna Holmqvist, Nils Berginström, Marika C Möller.

**Supervision:** Marika C Möller.

**Visualization:** Love Engström Nordin.

**Writing – original draft:** Anna Holmqvist.

**Writing – review & editing:** Anna Holmqvist, Love Engström Nordin, Nils Berginström, Monika Löfgren, Lars Nyberg, Britt-Marie Stålnacke, Marika C Möller.

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
