## [Decision Letter · Decision Letter 0]

8 Jul 2025

PONE-D-25-24612
Cognitive fatigability and neuronal correlates in chronic pain – a cross-sectional fMRI study
PLOS ONE

Dear Dr. Holmqvist,

Thank you for submitting your manuscript to PLOS ONE. After careful consideration, we feel that it has merit but does not fully meet PLOS ONE’s publication criteria as it currently stands. Therefore, we invite you to submit a revised version of the manuscript that addresses the points raised during the review process.

The reviewer has raised important methodological concerns, with particular emphasis on statistical issues and fMRI data analysis procedures. Please address these comments thoroughly in your revised manuscript, providing detailed explanations and any necessary corrections to strengthen the analytical framework of your neuroimaging study.

We look forward to receiving your revised manuscript.

Kind regards,

Rei Akaishi

Academic Editor

PLOS ONE

“Stiftelsen Promobilia (no. A22056).

The Department of Clinical Sciences, Karolinska Institutet.

Regional agreement between Umeå University and Västerbotten County Council on cooperation in the field of Medicine, Odontology, and Health.”

3. In the online submission form you indicate that your data is not available for proprietary reasons and have provided a contact point for accessing this data. Please note that your current contact point is a co-author on this manuscript. According to our Data Policy, the contact point must not be an author on the manuscript and must be an institutional contact, ideally not an individual. Please revise your data statement to a non-author institutional point of contact, such as a data access or ethics committee, and send this to us via return email. Please also include contact information for the third party organization, and please include the full citation of where the data can be found.

Reviewers' comments:

Reviewer's Responses to Questions

**Comments to the Author**

1. Is the manuscript technically sound, and do the data support the conclusions?

Reviewer #1: Yes

Reviewer #2: Yes

2. Has the statistical analysis been performed appropriately and rigorously? 

Reviewer #1: Yes

Reviewer #2: Yes

3. Have the authors made all data underlying the findings in their manuscript fully available?

Reviewer #1: No

Reviewer #2: Yes

4. Is the manuscript presented in an intelligible fashion and written in standard English?

Reviewer #1: Yes

Reviewer #2: Yes

5. Review Comments to the Author

Reviewer #1: As the statistical reviewer I will focus on methods and reporting.

Major

1) some of the language on the findings, and the conviction with which the authors are taking about the findings, based on a p-value from a test including 46 subjects, needs to be moderated. e.g. "the study demonstrated.." more work is needed on this space and a larger sample to confirm findings. Overall, interpretation would need to move away from p-values and focus on effect sizes and future work, something which is not easy to do with the current analytical approach (as opposed to regression models).

2) there is no research statement followed - please adhere to the STROBE or another relevant research statement.

3) there is no mention of missing data, if any, and how that was dealt with. needs to be included at the end of the methods section.

4) limited generalisability is another concern, which the authors highlight (no action needed).

5) The control group had significantly higher education levels, which could influence cognitive performance and brain activation patterns. This should be discussed as a limitation.

6) The causal link between increased brain activation and compensatory mechanisms is tenuous. Alternative explanations like inefficiency or pain interference could be considered. The lack of time-dependent neural effects despite behavioural fatigability weakens the link between brain activity and cognitive fatigability. These interpretations should be presented more cautiously, especially given the small sample size and potential confounding factors.

Minor

1) "matched as closely as possible at the group level for age, and educational background". How? more details are needed.

2) The distinction between "cognitive fatigue" and "cognitive fatigability" should be clarified further,

Reviewer #2: Thank you for allowing me to review this manuscript.

The study is interesting and useful to understand relationship between chronic pain and fatigue. However, the manuscript is not well organized and there are several concerns.

1. The author suggested “When examining changes in BOLD fMRI signal between the different quartiles, we found no significant time effect”. Detailed information needs to be provided except for those in Table 3. Curves or other materials can be provided to describe the time effect.

2. The study emphasized the neural correlates of the fatigue in chronic pain, but the description of the results was very simple and many item was lost. For example, in Table 3, no P-values were provided. The abbreviations, i.e., IC, MNI, were not explained behind the table.

3. Description of the group differences of the functional connectivity is roughly. In the “Results, fMRI data” section, please provide the peak voxel coordinates, cluster size, p-value, t/z values of the brain regions showing significant between-group differences.

4. For Figure 1, please interpret the meaning of different colors, and add the color bar. For the group differences of the functional connectivity, which correction methods were used? What is the p values? Please clarify in both the manuscript, tables, and figure.

5. Is the fMRI results correlated with clinical measures? No results of correlation analysis were provided. Please added relative contents.

6. All tables are not three-line table.

7. All the abbreviations in the figures and tables need to be explained in the Figure legends or table annotations.

8. Table 1, 3rd line, the format of patient number is not correct. It is confusing for the number “100”. The correct format is “n(%)”. For example, 24(100%). Please revise it.

6. PLOS authors have the option to publish the peer review history of their article (what does this mean?). If published, this will include your full peer review and any attached files.

Reviewer #1: No

Reviewer #2: No

---

## [Author Response · Author response to Decision Letter 1]

21 Aug 2025

PONE-D-25-24612

Cognitive fatigability and neuronal correlates in chronic pain – a cross-sectional fMRI study

PLOS ONE

Dear editor,

Thank you for your assistance with the manuscript! Below, we address the requests/recommendations made by you and the reviewers.

Answers to the editor

Response: We have now revised the manuscript to meet PLOS ONE´s style requirements.

“Stiftelsen Promobilia (no. A22056).

The Department of Clinical Sciences, Karolinska Institutet.

Regional agreement between Umeå University and Västerbotten County Council on cooperation in the field of Medicine, Odontology, and Health.”

Please state what role the funders took in the study. If the funders had no role, please state:

Response: We have now stated that the funders had no role in study design, data collection and analysis, decision to publish, or preparation of the manuscript in the Cover letter.

3. In the online submission form you indicate that your data is not available for proprietary reasons and have provided a contact point for accessing this data. Please note that your current contact point is a co-author on this manuscript. According to our Data Policy, the contact point must not be an author on the manuscript and must be an institutional contact, ideally not an individual. Please revise your data statement to a non-author institutional point of contact, such as a data access or ethics committee, and send this to us via return email. Please also include contact information for the third party organization, and please include the full citation of where the data can be found.

Response: We have now revised our data statement to the ethics committee; registrator@etikprovning.se, telephone +46 010 475 08 00, and sent this information to you by e-mail, as requested.

Response: The ethics statement now only appear in the Methods section of the manuscript. Please see p 12, lines 227-229

Answers to reviewers

Reviewer #1:

Dear reviewer,

Thank you for reading the paper with care and scrutiny. Your comments and criticism are indeed of great relevance to enhance the overall quality of the paper. Replies to your comments are given below point by point. References to page and line numbers in the manuscript are given where applicable and are also marked in the revised manuscript.

As the statistical reviewer I will focus on methods and reporting.

Major

1) some of the language on the findings, and the conviction with which the authors are taking about the findings, based on a p-value from a test including 46 subjects, needs to be moderated. e.g. "the study demonstrated.." more work is needed on this space and a larger sample to confirm findings. Overall, interpretation would need to move away from p-values and focus on effect sizes and future work, something which is not easy to do with the current analytical approach (as opposed to regression models).

Response: We acknowledge your point but since our statistical methods do not include calculations of effect sizes we have now modified the interpretation of the results, emphasizing the limitations of the study and the need for future well- designed studies to confirm our findings as follows: p 3, line 43, 47 and 48 ; p 19, line 379 and 382-385: However, the lack of time-dependent neural effects despite behavioral fatigability weakens the link between brain activity and CF. It should be stated that this is a small explorative study, and that repeated well-designed studies that could confirm our preliminary findings are required.

2) there is no research statement followed - please adhere to the STROBE or another relevant research statement.

Response: Thank you for this relevant remark! We have followed the STROBE guidelines, which is now stated in the manuscript. Please see p 6 line 92 and 95: The study is presented according to the STROBE guidelines.

3) there is no mention of missing data, if any, and how that was dealt with. needs to be included at the end of the methods section.

Response: Thank you for pointing out that we missed providing this important information – it has now been added to the manuscript, please see Table legends, Table 1 and 2. Information about motion correction and exclusion limit is added to line 189-191 (p 10) in the Image post-processing section: Data with motion exceeding 2-3 mm cannot be reliably corrected with standard motion correction algorithms and can therefore cause misregistration. In the same section a sentence was added at the end explaining that no data was excluded for this reason, please see p 11 line 216-220: Separate contrast matrixes were constructed using GLM to search for within group time effects by comparing each quartile to all other quartiles. This analysis was performed in both directions for patients and controls respectively. Also, the time effect analyses were corrected for multiple comparisons using 10000 permutations with randomize and a Bonferroni corrected p-value. No data sets were excluded because of motion.

4) limited generalisability is another concern, which the authors highlight (no action needed).

Response: -

5) The control group had significantly higher education levels, which could influence cognitive performance and brain activation patterns. This should be discussed as a limitation.

Response: This is a valid point. The difference in educational levels between the groups is now discussed as a limitation. Please see p 18 line 362-363: Also, the control group had significantly higher levels of education, which may have influenced both cognitive performance and patterns of brain activation.

6) The causal link between increased brain activation and compensatory mechanisms is tenuous. Alternative explanations like inefficiency or pain interference could be considered. The lack of time-dependent neural effects despite behavioural fatigability weakens the link between brain activity and cognitive fatigability. These interpretations should be presented more cautiously, especially given the small sample size and potential confounding factors.

Response: We realize your point here and the weaknesses pointed out have now been addressed in the limitations section. Please see p 19 line 382-385: However, the lack of time-dependent neural effects despite behavioral fatigability weakens the link between brain activity and CF. It should be stated that this is a small explorative study, and that repeated well-designed studies that could confirm our preliminary findings are required.

Minor

1) "matched as closely as possible at the group level for age, and educational background". How? more details are needed.

Response: The matching procedure is now described in more detail, as suggested. Please see p 7 line 118-121: The controls were included in parallel with the patients, and care was taken to ensure that gender, age and length of education matched the patient group. However, it was difficult to recruit controls with a low level of education, resulting in an incomplete match in educational background.

2) The distinction between "cognitive fatigue" and "cognitive fatigability" should be clarified further,

Response: Thank you for this remark! The distinction is now clarified in the introduction section. Please see p 5 line 55-60: Experience of fatigue can be categorized into different subtypes, such as physical or mental, but also cognitive fatigue, the latter being conceived as the experience of fatigue resulting from cognitive effort (2). Cognitive fatigue has, however, not been well studied in patients with chronic pain (3). An aspect of cognitive fatigue is cognitive fatigability (CF), which means that cognitive performance deteriorates over time. Thus, CF can be assessed objectively as performance decrement on tasks requiring sustained mental effort (4, 5)., and line 64.

Reviewer #2:

Dear reviewer,

Thank you for reading the paper with care and scrutiny. Your comments and criticism are indeed of great relevance to enhance the overall quality of the paper. Replies to your comments are given below point by point. References to page and line numbers in the manuscript are given where applicable and are also marked in the revised manuscript.

1. The author suggested “When examining changes in BOLD fMRI signal between the different quartiles, we found no significant time effect”. Detailed information needs to be provided except for those in Table 3. Curves or other materials can be provided to describe the time effect.

Response: Thank you for pointing this out, it needs to be described in more detail. We performed regression analysis between the different time points of the fMRI data within each group using the FSL-tool dual regression. These regression analyses were performed in several different ways;

Q1 and Q2 vs Q3 and Q4, Q1 vs Q2, Q1 vs Q3, Q1 vs Q4, Q2 vs Q3, Q2 vs Q4, Q3 vs Q4 (Q for quartile).

Dual regression does one way analysis so all the above contrasts were also performed the other way around for both patients and controls to make sure all possible time effects could be seen in both groups. When reviewing the results from the analysis, a p-value threshold is set. When this threshold was set to 0.05 (5%) no significant results could be seen, therefore we cannot present anything from this. We have clarified the analysis in the Methods section / Image post processing and in the results section / fMRI data. Please see p 10 line 189-191: Data with motion exceeding 2-3 mm cannot be reliably corrected with standard motion correction algorithms and can therefore cause misregistration, p 11 line 216-220: Separate contrast matrixes were constructed using GLM to search for within group time effects by comparing each quartile to all other quartiles. This analysis was performed in both directions for patients and controls respectively. Also, the time effect analyses were corrected for multiple comparisons using 10000 permutations with randomize and a Bonferroni corrected p-value. No data sets were excluded because of motion, and Table 3.

To visualize the results, we have chosen to include a figure demonstrating fatigability as measured by PVT performance in this revised version. Please see Figure 1.

2. The study emphasized the neural correlates of the fatigue in chronic pain, but the description of the results was very simple and many item was lost. For example, in Table 3, no P-values were provided. The abbreviations, i.e., IC, MNI, were not explained behind the table.

Response: The final results from dual regression consist of p-value statistics. A p-value threshold is applied and every voxel from the statistical maps with a significance level below the set threshold value is cancelled. For all clusters in table 3 the p-value threshold 0.05 was chosen meaning that all clusters have a p-value of 5% or lower. We have tried to explain this better in the table text and added the abbreviations below the table. Within each cluster, each single voxel has its own specific p-value but we can´t present all of them in a reasonable table, therefore we only present the p-value threshold which we consider to be standard for this type of analysis. Please see Table 3.

3. Description of the group differences of the functional connectivity is roughly. In the “Results, fMRI data” section, please provide the peak voxel coordinates, cluster size, p-value, t/z values of the brain regions showing significant between-group differences.

Response: The MNI coordinates in table 3 represents the centre point of each cluster, this has now been explained in the table text. We have also explained the p-value threshold, see above. In the table we chose to only present the cluster volume for the purpose of easy interpretation. We have now also added the cluster size as number of acquired voxels. The z-statistics maps are generated by step 2 of 3 in dual regression. In the 3rd and final step, the z-statistics maps are used for randomize to avoid multiple comparisons in the group statistics, resulting in the final corrected p-value images. Hence, as described above, we present the p-value thresholded clusters. The z-value statistics is not corrected for multiple comparisons and should therefore not be presented in the table.

4. For Figure 1, please interpret the meaning of different colors, and add the color bar. For the group differences of the functional connectivity, which correction methods were used? What is the p values? Please clarify in both the manuscript, tables, and figure.

Response: The different colors in the figure are not a result of the statistics, but rather that the significant clusters have been saved as separate files. Thank you for noticing this mistake which may seem confusing. We have now presented an updated figure with only two colors, blue for patients and red for controls. Note, since we added a new figure, this is figure 2 in the current version.

5. Is the fMRI results correlated with clinical measures? No results of correlation analysis were provided. Please added relative contents.

Response: This is indeed a relevant remark. As we found group differences on the behavioral fatigability measures (PVT) we conducted a correlations between the PVT results and self-reported current pain, mentioned in the discussion section, please see p 18, line 367-368: However, we found no correlation between current pain intensity and PVT- performance.

However, we realize that this result was not included in the result section, and we have now added it, please see p 13 line 262-263: There were no significant correlations between current pain measurements (VAS pain) and the results of the PVT.

 Regarding correlations between current pain and fMRI results, ee do not think it is of relevance conducting a correlation between self-reported pain and the fMRI results, since there was no significant difference between patients and controls regarding fatigability as measured with fMRI.

6. All tables are not three-line table.

Response: Thank you for this remark; all tables are now three-line tables.

7. All the abbreviations in the figures and tables need to be explained in the Figure legends or table annotations.

Response: Thank you for noticing this! Abbreviations are now explained in every table and figure legends.

8. Table 1, 3rd line, the format of patient number is not correct. It is confusing for the number “100”. The correct format is “n(%)”. For example, 24(100%). Please revise it.

Response: Thank you for this remark! Table 1, line 3 has now been corrected.

---

## [Decision Letter · Decision Letter 1]

4 Sep 2025

Cognitive fatigability and neuronal correlates in chronic pain – a cross-sectional fMRI study

PONE-D-25-24612R1

Dear Dr. Holmqvist,

We’re pleased to inform you that your manuscript has been judged scientifically suitable for publication and will be formally accepted for publication once it meets all outstanding technical requirements.

Kind regards,

Rei Akaishi

Academic Editor

PLOS ONE

Additional Editor Comments (optional):

Your manuscript has been formally accepted for publication. Please follow the journal editors' instructions for the subsequent publication processes.

Reviewers' comments:

Reviewer's Responses to Questions

**Comments to the Author**

1. If the authors have adequately addressed your comments raised in a previous round of review and you feel that this manuscript is now acceptable for publication, you may indicate that here to bypass the “Comments to the Author” section, enter your conflict of interest statement in the “Confidential to Editor” section, and submit your "Accept" recommendation.

Reviewer #1: All comments have been addressed

Reviewer #2: All comments have been addressed

2. Is the manuscript technically sound, and do the data support the conclusions?

Reviewer #1: Yes

Reviewer #2: Yes

3. Has the statistical analysis been performed appropriately and rigorously? 

Reviewer #1: Yes

Reviewer #2: Yes

4. Have the authors made all data underlying the findings in their manuscript fully available?

Reviewer #1: Yes

Reviewer #2: Yes

5. Is the manuscript presented in an intelligible fashion and written in standard English?

Reviewer #1: Yes

Reviewer #2: Yes

6. Review Comments to the Author

Reviewer #1: I am satisfied with the authors' responses and the resulting changes to the paper. I have nothing further to add.

Reviewer #2: The authors have addressed all my concerns. All the comments have been replied. The tables and figures have been added and revised.

7. PLOS authors have the option to publish the peer review history of their article (what does this mean?). If published, this will include your full peer review and any attached files.

Reviewer #1: No

Reviewer #2: No

---

## [Editor Report · Acceptance letter]

PONE-D-25-24612R1

PLOS ONE

Dear Dr. Holmqvist,

I'm pleased to inform you that your manuscript has been deemed suitable for publication in PLOS ONE. Congratulations! Your manuscript is now being handed over to our production team.

Kind regards,

on behalf of

Dr. Rei Akaishi

Academic Editor

PLOS ONE